# Genetic Diversity of Common Olive (*Olea europaea* L.) Cultivars from Nikita Botanical Gardens Collection Revealed Using RAD-Seq Method

**DOI:** 10.3390/genes14071323

**Published:** 2023-06-23

**Authors:** Natalia Slobodova, Fedor Sharko, Maria Gladysheva-Azgari, Kristina Petrova, Sergey Tsiupka, Valentina Tsiupka, Eugenia Boulygina, Sergey Rastorguev, Svetlana Tsygankova

**Affiliations:** 1National Research Center “Kurchatov Institute”, Moscow 123182, Russia; 2Faculty of Biology and Biotechnology, HSE University, Moscow 101000, Russia; 3Research Center of Biotechnology of the Russian Academy of Sciences, Moscow 119071, Russia; 4Research Center for Medical Genetics, Moscow 115522, Russia; 5Nikita Botanical Gardens–National Scientific Centre of the Russian Academy of Sciences, Yalta 298648, Russia; 6Pirogov Russian National Research Medical University, Moscow 117997, Russia

**Keywords:** *Olea europaea* L. (olive tree), ddRAD sequencing, geographical distribution of cultivars, fixation index, identity by state

## Abstract

In different countries, interest in the commercial cultivation of the olive has recently greatly increased, which has led to the expansion of its range. The Crimean Peninsula is the northern limit of the common olive (*Olea europaea* L.) range. A unique collection of common olive’s cultivars and hybrids has been collected in the Nikitsky Botanical Gardens (NBG). The aim of this study was to assess the genetic diversity of 151 samples (total of several biological replicates of 46 olive cultivars including 29 introduced and 11 indigenous genotypes) using the ddRAD sequencing method. Structural analysis showed that the studied samples are divided into ten groups, each of which mainly includes cultivars of the same origin. Cultivars introduced to the Crimean Peninsula from different regions formed separate groups, while local cultivars joined different groups depending on their origin. Cultivars of Crimean origin contain admixtures of mainly Italian and Caucasian cultivars’ genotypes. Our study showed that the significant number of Crimean cultivars contains an admixture of the Italian cultivar “Coreggiolo”. Genetic analysis confirmed the synonymy for the cv. “Otur” and “Nikitskaya 2”, but not for the other four putative synonyms. Our results revealed the genetic diversity of the olive collection of NBG and provided references for future research studies, especially in selection studies for breeding programs.

## 1. Introduction

Common olive (*Olea europaea* L.) is one of the oldest cultivated plants, the history of domestication of which began in the east of the Mediterranean approximately five to seven thousand years ago [1,2,3]. Early domesticated forms probably spread during human migration from east to west and introgressed with local wild olives, which, in turn, gave rise to local cultivated forms as a result of selection [4,5,6]. Currently, olive trees occupy 10.3 million hectares of land around the world [7], mainly in the Mediterranean basin. Increased commercial interest outside the Mediterranean basin resulted in its range expansion to Japan, China, the USA, South America, and Australia [8,9].

The expansion of areas occupied by olive trees is associated with a high demand for olive oil, as well as success in creating new highly productive cultivars that can grow on the northern limit of the range [10]. Works on the creation of new olive cultivars are hampered by the lack of cultivar authentication, which resulted in conflicting scientific results [11]. According to G. Bartolini, there are more than 2600 cultivars in the world [12], and 250 of them are classified by the International Olive Oil Council as “commercial cultivars” [13]. At the same time, S. Duran mentions more than 1200 cultivars [14], and C. Breton–more than 2000 cultivars [4]. The FAO Plant Production and Protection Division estimates that the world’s olive germplasm collection contains more than 1200 different cultivars and more than 3600 synonyms [15,16], and many local cultivars and ecotypes contribute to this richness. Synonyms (different names for the same cultivar) and homonyms (the same name for different cultivars) are extremely common among countries that grow olives [17]. In addition, the occurrence of clonal mutations, which may or may not have a specific phenotypic expression, makes the characterization of olive cultivars a complex process requiring expertise in both morphological and genetic identification [18,19,20].

The assessment of genetic diversity and identification of cultivars is very relevant and is carried out in most genetic collections around the world. Such studies are widely represented in Spain [21,22], Italy [23], Egypt [24], Tunisia [25], and other countries and are the basis for obtaining reliable data in various interdisciplinary studies. The characteristics of olive cultivars have long been based only on morphological features [26,27]. Today, molecular markers have become the preferred tool for crop varietal identification and studies on genetic diversity and population structure [28]. Most molecular studies of *O. europaea* have been carried out using AFLP [29], ISSR [30], SSR [31,32,33,34], and SNP markers [35,36].

Despite the fact that the southern coast of Crimea (SCC) is the northern limit of the olive range, the coastal strip of the SCC provides good conditions for the cultivation of this crop [37]. Olive cultivars from different countries and regions grow here (Italy, Spain, France, Albania, and North African and Caucasus regions). The collection began shortly after the opening of the Nikitsky Botanical Gardens (NBG) in 1812, while the history of growing olives in the south coast of Crimea dates back more than a century. At the moment, the NBG has collected various cultivars (introduced from different regions and autochthonous) bred on the Crimean coastline. At the same time, some cultivars were obtained via open pollination of known cultivars, or were discovered during plantation exploration on the Crimean Peninsula. Now, the NBG collection includes more than 260 cultivars of olives and about 2000 hybrid seedlings, including hybrid forms of the Botanical Gardens selection.

The genetic analysis of olive cultivars from the NBG collection is being carried out for the first time. For this purpose, high-resolution SNP markers were used, allowing a thorough characterization of the genetic structure of populations, assessment of variability, and genotyping of available cultivars. We applied restriction site-associated DNA sequencing (RAD-sequencing), which has the advantages of next-generation sequencing (NGS) technology for population studies at a relatively low cost [38]. To date, several modifications of this method have been developed, including ddRAD sequencing, which allows multiplexing large-scale samples since it contains the step of including a four-index sequence [39]. The aim of the study was to assess the genetic diversity of common olive cultivars growing on the SCC. Our results shed light on the origin of these cultivars, allowing them to be identified and defined synonyms and homonyms. The authors believe that the results obtained will become the basis for further interdisciplinary research on the collection of olives from the south coast of the Crimea.

## 2. Materials and Methods

### 2.1. Plant Material and DNA Extraction

Perennial plants of the olive cultivars growing in the collection plots of the Nikita Botanical Gardens—National Scientific Center of the RAS (NBG, Yalta) in the same climatic and soil conditions were used for the studies. The locations of the sampling and experimentation sites are indicated by the (44°50′66″ N 34°23′71″ E, 20–50 m above sea level) GPS coordinates. The area has a subtropical climate with dry and hot summers and humid winters, with rainfall mainly concentrated in the autumn and winter seasons. Annual average sunshine is 2285 h, and precipitation is 188 mm between May and September, and 595 mm for the whole year.

The absolute maximum temperature is 39.0 °C, and the absolute minimum is −14.6 °C. The average annual temperature is +12.4 °C, and the average annual air humidity is 67%. The soil is brown, weakly calcareous, and has heavy loamy clay shales and limestone [40].

In this study, 151 samples (total of several biological replicates of 46 olive cultivars including 29 introduced and 11 indigenous genotypes) from the NBG collection were collected and analyzed. A total of 29 cultivars were represented by olive introduced to the territory of the Crimean Peninsula (from Italy, Spain, Albania, France, Azerbaijan, Dalmatia, Algeria, and the Caucasus), while the remaining 17 were local autochthonous cultivars. Selected cultivars have the most accurate morphoanatomical description; they differ in their habitats and are planned for use in consequent breeding studies by NBG. Accession information, including cultivar name, country of origin, and pedigree, is provided in Appendix A. For each cultivar, 2–4 biological replicates (trees) were taken. Total genomic DNA was extracted from fresh young leaves according to the method of Lo Piccolo et al., 2012 [41], with minor modifications. The quality and quantity of DNA was assessed spectrophotometrically on a Nanodrop 1000 device (Thermo Scientific, Waltham, MA, USA) and using a Qubit fluorometer (Invitrogen, Waltham, MA, USA) with the Qubit™ dsDNA BR Assay Kit.

### 2.2. RAD Libraries Preparation and Sequencing

The genomic DNA of 151 samples was subjected to double cleavage with restriction endonucleases MspI and PstI. The adapters were ligated (complementary to the restriction sites on one side and containing a unique barcode) at the same time at 30 °C for 3 h. Then, samples with different combinations of internal barcodes were pooled equimolarly into pools of 12 samples, and each pool was purified from the remaining reaction components using Agencourt AMPure XP magnetic beads (Beckman Coulter, Brea, CA, USA). For each pool, a target distribution of 600 to 800 nucleotide fragments was collected on a BluePippin instrument (Sage Science Inc., Beverly, MA, USA). Next, PCR was performed to enrich the libraries and to attach external unique TruSeq indices with consequent purification on AMPureXP beads [42]. The quality and concentration of the resulting libraries were checked using a Qubit 2.0 fluorometer (Invitrogen, Carlsbad, CA, USA) and a DNA Bioanalyzer (Agilent Technologies, Santa Clara, CA, USA). The ddRAD libraries were sequenced using an S1 Illumina NovaSeq6000 flow cell (Illumina, Hayward, CA, USA) with paired-end reading (2 × 150 bp long).

### 2.3. Data Analysis

To demultiplex the reads of 151 samples according to the given barcodes, the process_radtags program (Version 2.64) [43] was used with the MspI and PstI restriction parameters. Analysis of the quality of the obtained nucleotide sequences was carried out using the FASTQC program (https://www.bioinformatics.babraham.ac.uk/projects/fastqc/ accessed on 26 February 2023). Further, from the original readings, filtering by the quality and length of the sequence was performed using the Trim Galore software package (https://www.bioinformatics.babraham.ac.uk/projects/trim_galore/ accessed on 26 February 2023) (Version 0.6.5). To align reads to the reference genome of *O. europaea* var. *sylvestris* (GCF_002742605.1) and to search for variable loci, bowtie2 and gstacks [44] were used. The loci were filtered by the DP and GQ parameters using the R package vcfR [45]; poorly covered and oversaturated loci were removed based on the normal distribution indicators. Clustering of the samples was performed using the construction of a dendrogram using the NJ method (Neighbor-Joining method) using the poppr package [46]. The principal component analysis (PCA) was performed in the R environment using the dartR package [47], in which the Fst and identity by descent (IBD) statistics were also calculated, and the discriminant analysis of principal components (DAPC) was performed using the adegenet package [48]. To analyze the structure of the population, parametric methods were used, which were implemented in the ADMIXTURE v. 1.3.0 [49] using the maximum likelihood method. A cross-validation procedure was used to select the best K parameter, which describes the number of subpopulations in the total population.

## 3. Results

The sequencing of 151 common olive samples was performed with 226,627,595 paired reads in total and an average of 1,500,845 per sample (369,181 to 4,302,146) with a mean base sequence quality of 36. In total, after filtering for quality, 191,160,376 (84.35%) reads were mapped to the reference genome of *O. europaea* var. *sylvestris*, and the distribution of mapping percentage between accessions ranged from 59.97% to 89.33% with a mean MAPQ = 23.0585. The total number of variable loci was 695.385. A total of 110,152 SNP were left after DP and GQ filtering for population and subsequent analyzes, each of which was present in at least 95% of the samples. To assess the structure of genetic diversity within the entire common olive collection, principal component analysis (PCA) was used. Its results are presented in Figure 1.

Of the introduced cultivars, Italian ones formed five separate groups (marked in green in Figure 1), and Albanian ones were divided into three groups (marked in red in Figure 1). The cultivars of the NBG (blue in Figure 1) were divided into five groups. Caucasian cultivars formed separate groups as well; according to the PCA analysis, we identified five different ones.

The Bayesian clustering approach was applied to determine the genetic structure of the sampling [50]. Analysis of the RAD-sequencing data by cross-validation showed that the studied samples can be divided into 10 groups. The optimal number of clusters was selected using the k-mean algorithm [51]. The value of K is marked on the abscissa, and the cross-validation error, CV-error, is marked on the ordinate. The model shows the best values at K = 10 (Appendix A (Determination of the best K number)), so sampling was divided into 10 groups (V) (Figure 2). In this case, the geographical origin of the cultivars was taken into account. For group classification, we considered a genotype uniquely assigned to a group when its admixture ratio was >80% (Q > 0.8), as described previously [6].

As a result of structural analysis (Figure 2), olive cultivars of Albanian origin formed two clusters (V3 and V6). V6 is represented only by Albanian cultivars and corresponds to the same PCA group. In the V3 cluster, in addition to two Albanian cultivars, there are two Italian cultivars and one of the NBG selection. According to the PCA, these Albanian cultivars were separated into two different groups, one of which included the Italian cultivars from V3, and the second consisted of the “D’Espagne” cultivar and the Albanian “Pulazeqin”.

All Italian cultivars were categorized into mixed clusters with cultivars from other geographical regions: V3, V5, V7, V8, and V10. In this case, five groups were also formed, but the composition of these groups was somewhat different from PCA. Caucasian cultivars were also distributed over several clusters. Cultivars of the NBG origin joined six different groups: V1, V2, V3, V7, V8, and V9. Some cultivars were grouped with ones from the Caucasus, and the other half with cultivars from Italy and Albania. Among the Crimean cultivars, some of them contained contributions from several ancestral groups (“Krymskaya Prevoskhodnaya” and “Yubileinaya”), and cultivars in the genome of which only one ancestral form (one color segment) was detected (for example, “Miskhorskaya”, “Nikitskaya”, and “Nikitskaya 5”).

We performed cluster analysis based on Euclidean distances using a filtered set of SNPs obtained by RAD-sequencing using the Neighbor-Joining (NJ) method (Figure 3). Cluster analysis reflected the structure of the population, similar to that obtained by the clustering method and PCA. Distinct olive cultivars occupied positions on the tree determined by the contribution of ancestral forms. Cultivars, in the genome of which a mixture of ancestral forms was found, occupied an indefinite position, for example, “Krymskaya Prevoskhodnaya”, “Pulazeqin”, and “Ascolano”.

As a result of the discriminatory analysis of principal components (DAPC), the analyzed sample of olives was divided into nine groups (Figure 4), which is quite consistent with the division based on structural analysis, where ten groups were identified. The exception was the “Tavlinskaya” cultivar of Caucasian origin, which, according to the results of structural analysis, formed a separate cluster. Among all nine clusters, the cluster of Albanian cultivars, as well as cultivars from Algeria and France, were grouped separately from the rest of the groups, which probably indicates a distant relationship with other groups. The Italian and Caucasian cultivars were clearly divided into two separate ellipses, which also indicates their difference both from each other and from the Algerian, French, and Albanian cultivars. One Spanish cultivar entered the group of Italian ones, and “Dalmatica” (former Yugoslavia) entered the group of Caucasian ones. Crimean cultivars (the NBG selection) form their own separate group, which overlaps with the Italian and Caucasian groups, which suggests close genetic links between these samples (Figure 4). This assumption is also confirmed by the results of structural analysis (Figure 2); as can be seen from the histogram, in the genomes of the Crimean cultivars, there is a genetic component of Italian and Caucasian ancestral forms.

For the clusters that were formed by the sample of olive cultivars analyzed in the study based on the results of structural analysis, the fixation index (Fst) for the clusters was calculated (Table 1). The highest differentiation was found between the V4 cluster (Fst 0.265–0.374) and other groups. This cluster is represented by one cultivar “Tavlinskaya”.

In addition, fixation indices (Fst) were determined in pairs for all analyzed cultivars (Appendix A (Fst for all analyzed cultivars)), and 11 pairs showed the lowest values (Table 2). Also, the relationships between *O. europaea* cultivars in this study were further investigated by evaluating the common allele values of identity by state (IBS) and identity by descent (IBD); the results are shown in Appendix A.

Based on the paired values of the IBD coefficients calculated for all cultivars of olives, a heat map was obtained (Figure 5). This map allows us to assess the common origin of cultivars from the analyzed collection. Italian cultivars “Piangente” and “Razzo”, Albanian “Bidza” and “Vajsi i Peqinit”, as well as “Dalmatica” and “Tiflisskaya” showed the highest IBD coefficients.

## 4. Discussion

For successful breeding and obtaining new cultivars, it is necessary to understand the genetic structure of the existing collection, adapted to specific growing conditions. The olive collection of the Nikitsky Botanical Gardens occupies the northernmost limits of the range. The absolute minimum temperature here reaches −14.6 degrees Celsius, with an average January temperature of +3.1 degrees Celsius [40].

We studied the genetic diversity and structure of the common olive population, consisting of local and introduced cultivated olives. Several replicates for each cultivar were included in the study in order to neutralize the influence of intravarietal differences and confirm varietal identity.

Analysis of the SNP markers of the NBG olive collection made it possible to divide it into 10 groups. At the same time, most of the groups included cultivars characterized by a common origin, and also contained admixture cultivars close to them. Only one group—V4—consisted of one cultivar, “Tavlinskaya”, which had no admixtures and was introduced to the NBG from the Caucasus from the Artvinsky olive nursery of the Batumi region in 1910. The Fst index, which describes the genetic differentiation between subpopulations, showed the highest values for this cultivar. In terms of phenotype, the cultivar “Tavlinskaya” has a higher oil content compared to other cultivars of the garden collection [52], which, together with genetic characteristics, makes it a promising cultivar for olive breeding in the NBG.

According to the results of the discriminant analysis, the distribution of olive samples on the plot correlates with geographical origin, similar to the data obtained by other authors [53]. Olive cultivars were introduced to the Crimean Peninsula from different growing regions, and many have formed clearly distinct groups. Autochthonous cultivars on the plot also formed their own group, which, on the contrary, forms numerous intersections with groups of imported cultivars, mainly of Italian and Caucasian origin.

The analysis of the population structure made it possible to study in more detail the genetic interactions between all samples. “Kolhoznitsa”, “Obilnaya”, “Chernaya Rannyaya”, “Nikitskaya 37”, and “Skorospelaya” were obtained by open pollination of the Italian cultivar “Coreggiolo” [54]. Based on the analysis of the population structure, it was confirmed that they have a significant admixture of the “Coreggiolo” in their genotypes (Figure 2). This landrace Italian cultivar was used for breeding as it had a fairly high oil content and productivity. In addition, in the genotypes of the cultivars “Nikitskaya 29” and “Nikitskaya 35”, found during the examination of plantations on the southern coast of Crimea, a significant contribution of “Coreggiolo” was also found.

The genotype of the “Nikitskaya 33” olive has admixtures of other Italian cultivars and the Albanian cultivar (Figure 2). For two cultivars (“Yubileinaya” and “Krymskaya Prevoskhodnaya”) which were found during the examination of plantations, it is not possible to unambiguously identify the parental forms that made the main contribution to the origin. Structural analysis revealed admixtures of both Caucasian and Italian cultivars. According to the pomological data, the cultivar “Yubileinaya” was obtained by selecting seedlings of the cultivar “Rannyaya”, which originated from the open pollination of “Coreggiolo” [54,55]. Genetic analysis showed the presence of an admixture of the “Coreggiolo” genotype in this cultivar, in addition to four others. “Krymskaya Prevoskhodnaya”, according to existing data, is a cultivar from the open pollination of the cultivar “Prevoskhodnaya”, which is a synonym for “Gorvala” [56]. According to our data, “Krymskaya Prevoskhodnaya” indeed contains a significant admixture of the “Gorvala” genotype (Figure 2).

The spread of cultivated plants is often associated with human migrations, while cultivars of olives could be propagated by cuttings, distributed throughout the Crimean Peninsula, and renamed regardless of the name of the original cultivar. Phenotypic descriptions and studies based on a small number of genetic markers such as SSR and AFLP [57,58] do not unequivocally clarify existing genetic relationships and may not always identify synonyms. “Nikitskaya 5”, “Nikitskaya 6”, and “Nikitskaya 2” were found during the examination of plantations of the Crimean Peninsula. According to the phenotype, they are considered synonyms of some Caucasian cultivars, also present in the NBG collection [54,59].

“Nikitskaya 2” is presumably a synonym for “Otur”, imported from the Caucasus. Genetic analysis showed that “Nikitskaya 2” and “Otur” have a high similarity of genotypes. The IBS value was more than 0.95 (0.97) on the phylogenetic tree these cultivars, represented by several biological replicates, formed a separate branch (Figure 4). The differentiation coefficient also showed low values (Fst 0.011). The data obtained allow us to confirm their synonymy.

“Nikitskaya 5” and its putative synonym “Tolgomskaya” are characterized by the same phenotypic manifestations of traits (Appendix A), but they have noticeable differences in genotypic characteristics: Fst (0.331) and IBS (0.88), which does not allow us to confirm the initial assumption of their synonymy. “Nikitskaya 6” and its putative synonym “Tossiyskaya” differ not only in the results of genetic analysis: IBS (0.87) and Fst (0.287), but also in terms of phenotype. These cultivars are located on different branches of the phylogenetic tree (Figure 4), and according to the results of structural analysis, they belong to different genetic groups (Figure 2), but it should be noted that “Nikitskaya 6” has a small admixture of the genotype characteristic of “Tossiyskaya” (Figure 2). At the same time, the analysis of the genomes of “Nikitskaya 5” and “Nikitskaya 6” revealed that they have a very high similarity with each other (Table 2), as well as with the Italian cultivar “Della Madonna” (Figure 2 and Figure 4, Table 2). These three cultivars form a cluster on the tree and are in the same genetic group according to the results of the structural analysis of the population. The data obtained can probably be explained by the genetic relationship of these cultivars.

According to the results of the structural analysis of the population and phylogenetic analysis, a separate subgroup was formed by two cultivars found during the examination of the plantations of the peninsula (“Nikitskaya” and “Miskhorskaya”), the Caucasian cultivar “Gorvala”, imported in 1902 from the Artvinsky oilseed nursery, and an olive tree growing on the territory of the garden which is more than 1000 years old (“ancient olive”) [37]. It is noteworthy that all four cultivars showed a very high genetic homogeneity (Table 2), while the phenotypes of these cultivars do not match. It is possible that “Nikitskaya” and “Miskhorskaya” were vegetatively obtained from the “ancient olive” and spread by humans on the territory of the peninsula. The origin of the Gorvala cultivar, brought from the Caucasus, has not been precisely established, but the similarity of genotypes (Figure 2) also suggests the presence of a common ancestor, and phenotypic differences arose due to the difference in growing conditions. D’Agostino et al. [53] noted that cultivars that are genetically close to each other may be phenotypically different. Differences in light, altitude, soil composition, and water availability can greatly influence the physiological and morphological aspects of olive plants [60,61,62,63]. Adaptation to local conditions may explain the differences between accessions and may be of great interest for olive breeding.

Three pairs of cultivars from our study showed the highest IBD ratios: 1.03, 0.70, and 0.91. These are landraces, once introduced to the territory of the peninsula from Italy (“Piangente” and “Razzo”) and Albania (“Bidza” and “Vajsi i Peqinit”), as well as from Dalmatia and the Caucasus (“Dalmatica” and “Tiflisskaya”). The close relationship of Albanian cultivars is confirmed by the studies of A. Dervishi, indicating their autochthonous origin. In addition, as noted by A. Belaj and G. Besnard, there is a high genetic similarity between cultivars from the same or close geographical area [64,65].

As a result of our work, the genetic characteristics of common olive cultivars from the unique collection of the Nikitsky Botanical Gardens were obtained. The ddRAD sequencing method has proven to be effective for this kind of research. A case of synonymy was confirmed for two olive cultivars, while for others that were considered synonymous in phenotype, these data were not confirmed. In such cases, in our opinion, it is necessary to further refine the phenotypic data.

The results of genotyping obtained in our work provide valuable information and will become the basis for further extended studies on the diversity and varietal characterization of the common olive collection in order to develop breeding programs and obtain new cultivars with improved properties, taking into account the genetic characteristics of the existing ones.

## Figures and Tables

**Figure 1 genes-14-01323-f001:**
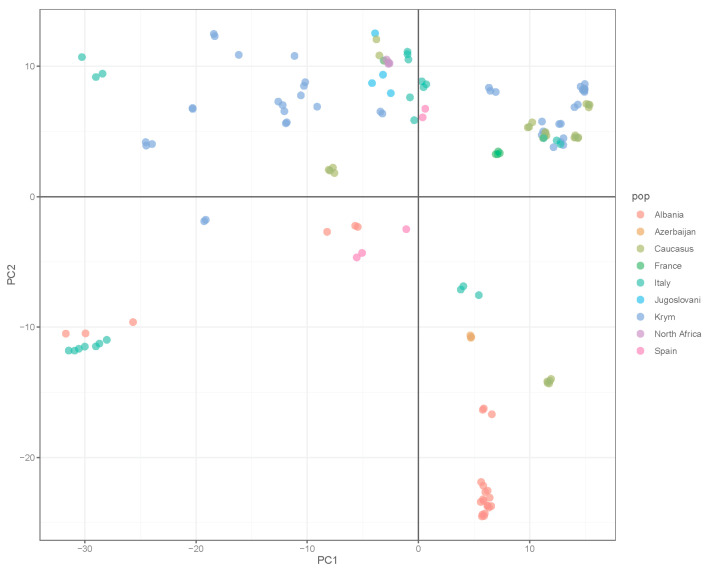
General PCA plot for 151 samples (46 common olive cultivars) from the NBG collection. Each cultivar is represented by several biological replicates. The color indicates the cultivar’s origin. PCA plots for each cluster and the general plot with cultivars’ names (with replicates numbered _1, _2, etc.) are represented in Appendix A (PCA plots for 151 samples from the NBG collection).

**Figure 2 genes-14-01323-f002:**
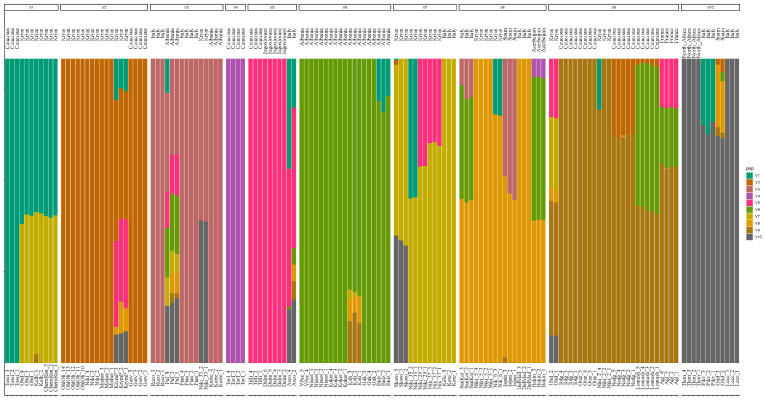
Barplot describing the genetic structure of the sample using the Bayesian approach. The population was divided into ten (K = 10) groups according to the most informative value of K. Colors indicate belonging to groups identified by Bayesian analysis. Each sample is represented by a thin vertical line that is divided into colored segments. Each cultivar is represented by 2–4 biological replicates. Each color represents one gene pool, and the length of the colored segment shows the estimated proportion of ancestors in that gene pool. Barplots with different K numbers are represented in Appendix A (Barplots describing the genetic structure with different K numbers).

**Figure 3 genes-14-01323-f003:**
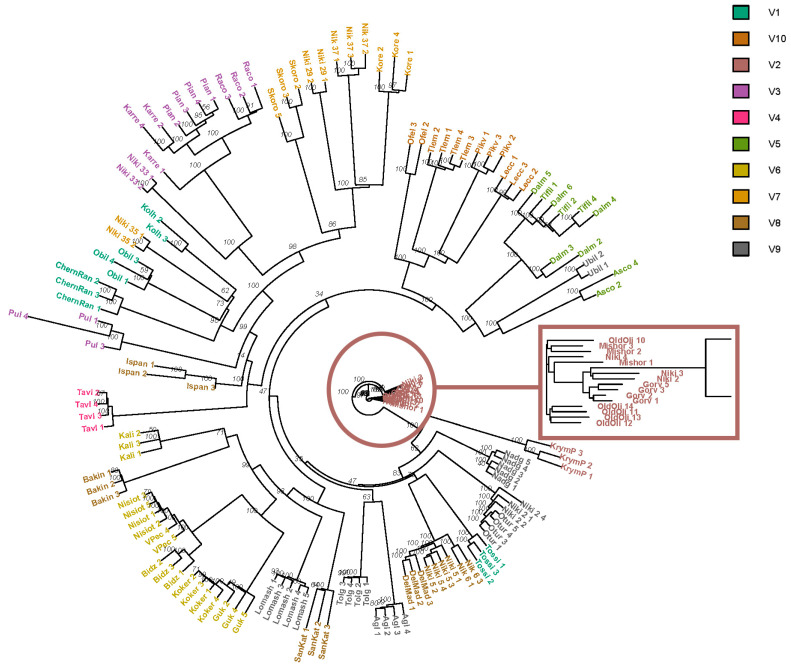
Dendrogram of cultivars constructed using the Neighbor-Joining (NJ) method based on the analysis of SNPs obtained via RAD sequencing. V1–V9: groups corresponding to clusters obtained as a result of structural analysis. Each cultivar is represented by several biological replicates. In a separate square, the cluster formed by the cv. “Nikitskaya”, “Miskhorskaya”, “Gorvala”, and “Old Olive Tree” is shown.

**Figure 4 genes-14-01323-f004:**
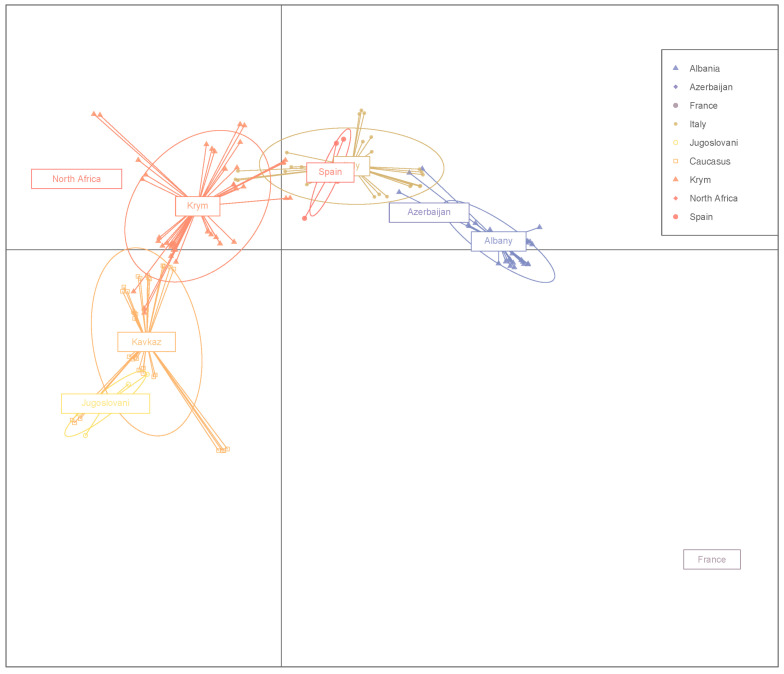
Principal component discriminant analysis (DAPC). The analysis displays the total genetic diversity of 151 accessions (46 cultivars) of the olive.

**Figure 5 genes-14-01323-f005:**
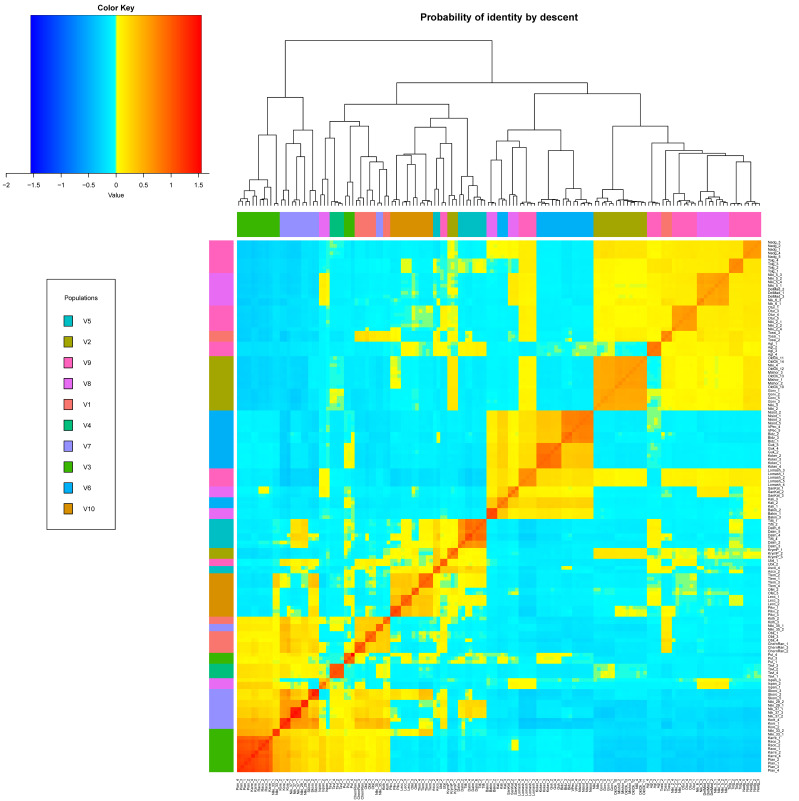
Heat map obtained from paired values of the IBD coefficient. Populations V1–V10: calculated on the basis of K-means clustering. Colors from blue to red show the degree of relationship with a common ancestor.

**Table 1 genes-14-01323-t001:** Matrix showing Fst values between clusters.

	V5	V2	V9	V8	V1	V4	V7	V3	V6	V10
**V5**	0									
**V2**	0.311	0								
**V9**	0.223	0.155	0							
**V8**	0.237	0.196	0.09	0						
**V1**	0.237	0.248	0.155	0.167	0					
**V4**	0.374	0.372	0.285	0.295	0.303	0				
**V7**	0.235	0.312	0.242	0.238	0.144	0.265	0			
**V3**	0.285	0.334	0.262	0.241	0.216	0.265	0.167	0		
**V6**	0.32	0.303	0.19	0.2	0.278	0.391	0.31	0.283	0	
**V10**	0.229	0.253	0.188	0.202	0.205	0.314	0.209	0.253	0.278	0

**Table 2 genes-14-01323-t002:** List of pairs of cultivars with common allele-by-state (IBS) values > 0.95 and pairwise Fst values between individual cultivars.

Cultivar1	Cultivar2	Fst	IBS
Nikitskaya 2	Otur	0.011	0.974
Nikitskaya 6	Nikitskaya 5	0.005	0.973
Nikitskaya 5	Della Madonna	0.027	0.980
Nikitskaya 6	Della Madonna	0.051	0.967
Nikitskaya	Miskhorskaya	0.019	0.969
Nikitskaya	Gorvala	0.045	0.983
“Old Olive Tree”	Miskhorskaya	0.006	0.976
“Old Olive Tree”	Gorvala	0.045	0.973
Gorvala	Miskhorskaya	0.049	0.971
Razzo	Piangente	0.073	0.962

## Data Availability

Raw data from our project were deposited in the NCBI SRA database under Bioproject ID PRJNA973272. Table with the reads number per sample attached as Appendix A (Table with the reads number per sample), reference genome: *Olea europaea* (GCF_002742605.1_O_europaea_v1/).

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
