# Peer review of "Genetic Diversity of Common Olive (Olea europaea L.) Cultivars from Nikita Botanical Gardens Collection Revealed Using RAD-Seq Method"

_genes, 2023, doi:10.3390/genes14071323_

Round 1

Reviewer 1 Report

The paper does a good job in clustering the olive cultivars from the extensive cultivar collection at NBG. The approaches they used (ddRAD and the analyses) are pretty standard and they are used and described adequately. I only have a few very minor corrections to fix typos or improve the presentation.

The paper comes across as somewhat descriptive. Just a suggestion (not a request - up to the authors whether to take this on board): it may be possible to add a bit of evolutionary perspective to the paper by discussing the results in the context of olive tree domestication. If anything is known about that, perhaps it is worth adding a paragraph on olive tree domestication to the introduction to explain what is known + some text to the discussion to link the findings of this study to the wider picture of domestication of this species.

It is mentioned that NBG collection includes 260 cultivars, but only 46 are analysed in this study. Would it be possible to add a few sentences clarifying the criteria that were used to select the subset of cultivars to be analysed?

Line 19 and elsewhere: it would be more informative to say that you analysed 151 samples from 46 cultivars. Otherwise it sounds as only 46 samples were analysed.

Line 172 "The total number of variable loci was 695.385" Perhaps it should be "695,385"?

On Fig1 and Fig4 "Albany" should be "Albania" and "Kavkaz" should be "Caucasus"

Line 181: Unclear whether the replicates of the same cultivar are shows separately (and numbered _1, _2 etc) or they are pooled into a single point and the number next to name shows the number of replicates pooled.

Fig1: The names and lines on the plot make it so cluttered that it is almost impossible to see the actual points and tell whether there are 46 or 151 points shown. I suggest to have a plot without names in the main text and show the plot with names in supplementary.

Line 190: Please add reference after "optimal number of clusters was se- 189 lected using the k-mean algorithm"

Line 188 or 189: Please add reference to Figure 2

Line 197: Please change " a sampling " to "the sample"

Line 295: Change "high" to "higher"

Reviewer 2 Report

The MS entitled 'Genetic diversity of common olive (Olea europaea L.) cultivars from Nikita Botanical Gardens collection revealed using RAD-Seq method' explore diversity and population structure of the common olive which use RAD-seq of 46 olive cultivars. This study provides basic study for Oliver breeding and molecular studies. However, I have several concerns which need to be addressed:

Introduction:

1.the Introduction part is too long, the cultivation history and the importance of oliver should combined into one paragraph. The population and genetic study should be combined into one. The introduction of your materials and your studies should be collapsed into two.

Results

1. sequencing data analysis part is lacking of content. Author should add more information about base number, mapping quality, SNP numbers and described it. Otherwise, this section is too short to be a single section.

2. The PCA plot is good way to show the cluster of samples. However, it’s kind of hard to find the different groups of Italian, Albanian, NBG and Caucasian from the whole PCA plot. I suggest that make some subplots of PCA to show these groups separately which can make reader more clearly understand the results. The whole PCA can be reserved as the whole scenario of all samples.

3. Although figure 2 can provide a very good explanation of your population structure. However, the quality of this figure is poor. I cannot read the top and bottom words of the figure very clearly. Further, maybe author can jointly show different k from 7-10 and other K in Supplementary figure.

The English of this MS is good and does not require significant modifications.
